# Effect of Dimethyloxalylglycine on Stem Cells Osteogenic Differentiation and Bone Tissue Regeneration—A Systematic Review

**DOI:** 10.3390/ijms25073879

**Published:** 2024-03-30

**Authors:** Qiannan Dong, Xiuzhi Fei, Hengwei Zhang, Ximei Zhu, Jianping Ruan

**Affiliations:** 1Key Laboratory of Shaanxi Province for Craniofacial Precision Medicine Research, College of Stomatology, Xi’an Jiaotong University, Xi’an 710000, China; 2Center of Oral Public Health, College of Stomatology, Xi’an Jiaotong University, Xi’an 710000, China

**Keywords:** bone regeneration, angiogenesis, hypoxia-inducible factor-1α, dimethyloxalylglycine, stem cells

## Abstract

Dimethyloxalylglycine (DMOG) has been found to stimulate osteogenesis and angiogenesis of stem cells, promoting neo-angiogenesis in bone tissue regeneration. In this review, we conducted a comprehensive search of the literature to investigate the effects of DMOG on osteogenesis and bone regeneration. We screened the studies based on specific inclusion criteria and extracted relevant information from both in vitro and in vivo experiments. The risk of bias in animal studies was evaluated using the SYRCLE tool. Out of the 174 studies retrieved, 34 studies met the inclusion criteria (34 studies were analyzed in vitro and 20 studies were analyzed in vivo). The findings of the included studies revealed that DMOG stimulated stem cells’ differentiation toward osteogenic, angiogenic, and chondrogenic lineages, leading to vascularized bone and cartilage regeneration. Addtionally, DMOG demonstrated therapeutic effects on bone loss caused by bone-related diseases. However, the culture environment in vitro is notably distinct from that in vivo, and the animal models used in vivo experiments differ significantly from humans. In summary, DMOG has the ability to enhance the osteogenic and angiogenic differentiation potential of stem cells, thereby improving bone regeneration in cases of bone defects. This highlights DMOG as a potential focus for research in the field of bone tissue regeneration engineering.

## 1. Introduction

Bone has limited regenerative ability, and various factors such as inflammation, trauma, tumors, and other causes can lead to oral and craniomaxillofacial bone defects that do not heal on their own. These defects have a significant impact on individuals’ daily lives and can even result in social problems, imposing a burden on the global economy and healthcare systems [1,2,3,4]. Currently, the ‘gold standard’ for clinical treatments is ‘autologous bone grafting’. However, its use is limited due to the scarcity of suitable donor sites [5], potential donor-site trauma [6], and complications such as bone nonunion and infection after transplantation [7]. Allogeneic bone grafting and the use of prosthetic materials also have drawbacks, including immune rejection [8] and post-implantation infections [9], which restrict their application in clinical practice. Moreover, regardless of the approach taken, whether it is autologous bone grafting, allogeneic bone grafting, or the use of prosthetic materials, the recovery period is lengthy, patients may experience discomfort, and the treatment can be expensive [10].

Stem cells exhibit robust self-renewal capabilities and the potential to differentiate into a variety of tissues and organs when exposed to specific conditions. They have been widely employed in the realm of bone tissue regeneration to address bone defects [11,12,13,14,15]. In the context of bone tissue regeneration, common sources for harvesting stem cells include bone marrow, adipose tissue, dental pulp tissue, and periodontal tissue, among others. Notably, bone marrow mesenchymal stem cells (BMSCs), adipose mesenchymal stem cells (ADSCs), dental pulp mesenchymal stem cells (DPSCs), and periodontal membrane ligament stem cells (PDLSCs) have been frequently utilized in this domain [16,17,18,19,20,21,22,23,24,25]. Furthermore, recent research has highlighted the potential of endometrial mesenchymal stem cells, which possess a strong capacity for proliferation, differentiation, and angiogenesis. Importantly, the harvesting process for these cells is relatively simple and non-invasive, making them a subject of considerable interest in the field of bone regeneration [26].

While significant progress has been made in stem cells research for regenerative medicine, there are still several factors to consider when applying stem cells in clinical treatment. These factors include age, disease, long-term expansion, and the unfavorable transplantation environment, which can impact the survival rate, regeneration, and differentiation characteristics of stem cells [27]. Various strategies have been explored to enhance the function of stem cells. These strategies encompass hypoxia treatment, small molecule drugs, bioactive factors, mechanical stimulation, and gene amplification. These may be promising approaches to improve the effectiveness of stem cells transplantation [28,29]. Additionally, combining biological materials with stem cells may also be an effective way to enhance the effectiveness of stem cells therapy. For instance, biopolymers and bioceramics have been shown to effectively promote stem cells osteogenesis due to their excellent mechanical properties, good drug release ability, and appropriate components. The internal porous structure of these materials also supports the formation of blood vessels, creating a favorable biological microenvironment for stem cells to repair bone defects [30].

The enhancement of cellular function in stem cells following hypoxia preculture is well known to be mediated by hypoxia-inducible factor (HIF). HIF serves as a crucial factor that cells activate to adjust to low oxygen levels, and it is the core regulator of the hypoxia response. It is part of the hypoxia-activated transcripts family and consists of three isoforms: HIF1, HIF2, and HIF3. HIF-1 is composed of two heterodimeric transcription factors, HIF-1α and HIF-1β. In a hypoxic environment, the dimer formed by HIF-1α and HIF-1β translocates to the cell nucleus and binds to the hypoxia response element (HRE). This binding influences the expression of various downstream target genes, leading to gene transcription and the regulation of cells proliferation, differentiation, migration, and homing, resulting in specific biological effects [31,32,33,34]. Research has highlighted the significant role of HIF-1α in bone tissue formation and its ability to promote stem cells differentiation into osteoblasts and vascular cells effectively [35,36,37]. Nevertheless, in a normal oxygen levels environment, prolyl hydroxylase (PHD) remains biologically active and triggers the von Hippel-Lindau (VHL) protein to bind to HIF-1α, deactivating it [38,39]. To enhance the expression of HIF-1α in cells under normal oxygen conditions, a viral transfection can be utilized. However, it is crucial to acknowledge that viral transfection carries the risk of cellular infection and tumor development [36,40,41].

Dimethyloxalylglycine (DMOG) is a pro-angiogenic small molecule drug with a chemical formula of C_6_H_9_NO_5_. DMOG is de-esterified in cells to form N-oxalylglycine (NOG), which inhibits the biological activity of PHD in a normal oxygen concentration environment and stabilizes the expression level of HIF-1α in cells under normoxia [42,43,44,45,46,47,48,49,50,51]. Its simple biochemical composition makes DMOG more stable and cost-effective, avoiding side effects like cells aging and reactive oxygen species (ROS) generation caused by cells cultured in a truly hypoxic environment. Research has demonstrated DMOG efficacy in treating bone turnover and bone loss induced by ovariectomy (OVX), and osteonecrosis of the femoral head (ONFH), as well as loss of alveolar bone density caused by obstructive sleep apnea-hypopnea syndrome (OSAHS) and nasal obstructions [52,53,54].

Acting as a dual regulator of osteogenesis and angiogenesis, HIF-1α effectively promotes the osteogenesis and angiogenesis of stem cells and plays an important role in bone tissue formation. Compared to other HIF-1α stabilizers, DMOG is better at maintaining the expression and secretion of HIF-1α under normoxia [55]. Therefore, the aim of this study is to review the existing literature on DMOG to explore its potential in enhancing stem cells function and improving treatment outcomes for bone deficiency-related diseases.

## 2. Materials and Methods

This systematic review follows the PRISMA statement [56] and formulates research questions using the PICO format. The population (P) includes in vitro or in vivo studies focusing on osteogenesis/angiogenesis and bone tissue regeneration. The intervention (I) involves osteogenic culture with DMOG-treated stem cells/DMOG treatment of bone defects, while the comparison (C) includes osteogenic culture with control stem cells (from untreated stem cells)/non-DMOG-treated treatment of bone defects. The outcome (O) assesses osteogenesis/angiogenesis in vitro and vascularized bone regeneration in vivo.

### 2.1. Search Strategy

The search strategy involved selecting articles from PubMed (National Library of Medicine, NCBI), Embase, Web of Science, and Scopus databases up to January 2024, using the terms: (Bone Regeneration OR Bone Tissue Engineering) AND Stem Cells AND (DMOG OR Dimethyloxalylglycine).

### 2.2. Eligibility Criteria

Articles were screened for inclusion in this study based on the following criteria: on the effects of DMOG in vitro related to osteogenic/angiogenic differentiation of stem cells; the effects of DMOG on bone tissue regeneration; the treatment of bone-related diseases using DMOG; or the assessment of in vitro/in vivo chondrogenesis or cartilage regeneration. The exclusion criteria included: articles that did not use stem cells as a study subject; using extracellular vehicles (EVs) as study subject; the purpose of the study was not relevant to bone tissue regeneration; non-mammalian studies; meeting abstract; reviews; and dissertations.

### 2.3. Study Selection

Retrieved records were screened for duplicates using the StArt tool for a systematic review (StArt is the latest technology for passing systematic reviews). Two researchers independently assessed articles’ eligibility. Initially, articles were assessed based on title, abstract, and then the full text.

### 2.4. Data Extraction

In this systematic evaluation, data extraction included both in vitro and in vivo information. Technical information extracted from the in vitro analyses included: source of stem cells, isolation methods, characterization identification, treatment groups, DMOG concentration, treatment duration, analysis reports, and results. For in vivo analyses, the extracted data included: types of animal model, bone defect models, treatment used (DMOG concentration, scaffolds, treatment duration), analysis reports, and results. Secondary results (signaling pathway analysis) were also extracted for the in vitro and vivo results. In addition, authors, titles, years of publication, journals, countries, and other information were also reported.

### 2.5. Risk of Bias Assessment

The risk of bias in animal studies was evaluated using the Systematic Review Center for Laboratory Animal Experiments (SYRCLE) tool [57].

## 3. Results and Discussion

### 3.1. Search Results

The initial database search retrieved a total of 174 studies. After removing duplicates, 44 out of 87 studies were screened by title and abstract. After a full-text review, a total of 34 studies [31,32,42,43,44,45,46,47,48,52,53,54,55,58,59,60,61,62,63,64,65,66,67,68,69,70,71,72,73,74,75,76,77,78] met the inclusion criteria and were included in this systematic evaluation. All 34 studies included in this review conducted in vitro analyses, and 20 studies were used for both in vitro and in vivo analyses (Figure 1).

### 3.2. Study Characteristics

The studies included in this review span from 2012 to 2024, with 2023 showing the highest publication activity. There has been a growing interest in treating stem cells with drugs or using drug-loaded bioactive scaffolds before applying the stem cells in disease treatment. This trend indicates that modulating the function of stem cells through the use of drugs and bioactive scaffolds is widely accepted in general science. Various journals, from general to specialized regenerative medicine journals, have published these studies. China has published the most research in this area. 

The focus of this paper is on examining the therapeutic impacts of the small molecule drug DMOG and bioactive materials containing DMOG on stem cells’ osteogenesis, angiogenesis, and bone defect diseases.

### 3.3. Isolation and Characterization of MSCs

Mesenchymal stem cells (MSCs) are a crucial source of stem cells in regenerative medicine research. BMSCs are the most commonly studied type, with human and rat sources being the primary focus. Human-derived BMSCs were used in nine studies [31,55,60,65,70,71,72,73,74], while rat-derived BMSCs were used in eight studies [32,43,44,47,54,58,64,77]. There is limited research on mouse-derived [63], pig-derived [75], and rabbit-derived [76] BMSCs. Apart from BMSCs, ADSCs have also been investigated for bone tissue regeneration. ADSCs from rats were utilized in three studies [42,66,67], human-derived ADSCs in two studies [48,69], and ADSCs from dogs [46] and rabbits [53] in one study each. Other studies explored the use of human PDLSCs [68] and porcine synovial-derived mesenchymal stem cells (SYN-MSCs) [78]. The most common method for cells extraction is tissue block enzyme digestion [31,42,43,44,46,47,48,53,66,68,69,75]. Flow cytometry (FCM) analysis was used in some studies to confirm the identity of MSCs, three-line differentiation was used to assess their ability to differentiate into osteogenic, adipogenic, and chondrogenic lineages. Nine studies performed FCM characterization of MSCs [31,42,45,46,47,48,55,68,70], while six studies examined their osteogenic, adipogenic, and chondrogenic differentiation characteristics [42,45,46,48,66,70]. Theses studies consistently reported that MSCs appeared as spindle or long spindle-like cells. The most frequently reported MSCs markers were CD29, CD44, CD73, CD90, CD104, CD105, and CD106, while lacking expression of CD14, CD34, CD45, CD80, and HLA-DR. MSCs demonstrated differentiation potential into osteogenic, adipogenic, and chondrogenic lineages. Furthermore, some studies explored the use of induced pluripotent stem cells (iPSCs) [45], and human umbilical vein endothelial cells (HUVECs) [59,61,62,63,67].

A summary of the MSCs isolation and characterization methods reported by the included studies can be found in Table 1.

### 3.4. In Vitro Studies

All studies included in this review conducted in vitro experiments. To ensure a fair comparative analysis, a group of stem cells without any treatment was present in all studies. For osteogenic differentiation analysis, the cells were typically treated with DMOG and cultured using osteogenic medium (OM). This was evaluated by measuring alkaline phosphatase (ALP) activity and alizarin red staining (ARS). The duration of osteogenic differentiation treatment varied among studies, with ALP activity assessed at 7 to 14 days in culture [42,43,54,59,60,61,62,63,64,67,68,73] and ARS culture lasting at least 21 days [42,43,54,59,61,62,63,67]. Osteogenic markers were generally observed between 3 to 21 days in culture [42,48,54,58,62,63,65,66,68,72,73,74]. For assessing angiogenic effects, some studies examined the expression of stem cells-associated angiogenic markers [32,42,43,44,45,46,47,48,54,58,62,65,66,68,69,70,72,73], while others assessed cells migration and tubule formation using DMOG on HUVECs [59,61,62,63,67]. It is worth noting that recognizing the crucial role of angiogenesis in achieving bone regeneration in vivo experiments on bone defects, certain studies focused on in vitro angiogenesis assessment rather than osteogenesis.

### 3.5. In Vivo Studies

A total of 20 in vivo experimental studies meeting the inclusion criteria were reviewed [42,43,44,45,47,52,53,54,58,59,61,62,66,68,71,72,73,75,76,77]. Among these, 13 studies utilized rats as the animal model [42,43,44,45,54,59,61,62,66,68,71,72,77], 5 studies used mice [52,61,62,75,76], and two studies used rabbits [53,58]. The defect models varied across the studies, with nine studies inducing skull defects [42,44,45,59,61,62,66,71,72], 2 studies creating mandibular defect models [43,68], and two studies utilizing rabbits for disease models (one for ONFH [53] and the other for femoral head defects [58]). Furthermore, 4 studies involved implanting scaffolds subcutaneously in mice [61,62,75,76], one study established a mouse OVX model [52], one study developed a rat OSAHS nasal obstructed model [54], and one study created a rat bilateral total knee joint cartilage defect model [77]. It is noteworthy that none of the studies on defect models explore bone defects in long bones or large animals.

The concentration of DMOG used to treat bone defects varied among studies due to different culture conditions. DMOG was incorporated into various scaffolds to investigate its impact on bone defects regeneration including gelatin sponge (GS) [43], β-tricalcium phosphate (β-TCP) [45], calcined bone calcium (CBC) scaffolds [58], nanoscale zeolitic imidazolate frameworks-8 (ZIF-8) and sodium alginate hydrogel (SA) [59], mesoporous silica nanospheres (MSN) [60,61,62], PCL grafts [61], nanofibrous gelatin-silica hybrid scaffold (GP) [62], natural wood elastic scaffolds [63], mesoporous bioactive glass (MBG) [64,65], sodium alginate-gelatin-β-tricalcium phosphate [66], nanocomposite hydrogel [67], poly (lactic-co-glycolic acid) fiber (PLGA) [68], nanoporous silica nanoparticles (NPSNPs) [69], MBG-dopedpoly(3-hydroxybutyrate-co-3-hydroxyhexanate) (PHMG) [72],calcium phosphate microparticle-s (CMPs) [73], metal-organic frameworks (MOFs) [74], polylactic acid scaffold (PLLA) [76], beaver nano hydrogel, chitosan (CS) and hydroxy propyl chitin hydrogel (HPCH) [77], and particulate-engineered scaffold [78]. The treatment duration ranged from 7 days to 18 weeks. Different articles also reported different time points that were analyzed.

To eliminate the potential influence of scaffolds and stem cells on bone defect regeneration, in vivo studies were categorized into a scaffold-only group. In cases where scaffolds were loaded with multiple drugs, additional non-DMOG drugs were included as controls.

Eighteen studies utilized microtomography for bone regeneration imaging and conducted quantitative analyses [42,43,44,45,52,53,54,58,59,61,62,66,68,72,73,75,76,77]. Seventeen studies presented histological findings [42,43,44,45,53,54,58,59,61,62,66,68,72,73,75,76,77]. Furthermore, 11 studies reported assessments of osteogenic and angiogenic markers through immunohistochemistry (IHC) staining: ALP, Runt related transcription factor 2 (Runx-2), type I collagen (Collagen-I), OCN (osteocalcin) analysis of osteogenesis; CD31, α-smooth muscle actin (α-SMA), vascular endothelial growth factor (VEGF), HIF-1α, and vascular endothelial growth factor receptor-2 (KDR) analysis of angiogenesis [42,43,44,45,53,54,58,66,68,72,73]. Four studies utilized immunofluorescence (IF) staining to examine osteogenic markers: Runx-2, OCN, Collagen-I, bone morphogenetic protein-2 (BMP-2), osteopontin (OPN), and the angiogenic markers CD90, CD31, CD34, HIF-1α, α-SMA, VEGF [59,61,62,68]. For a similar analysis, two studies used intraperitoneally injections with tetracycline, alizarin red, and calcein in a tricolor sequential fluorescent labeling method to visualize the studied bone regeneration progression [45,72]. Two studies evaluated osteoclast formation by tartrate-resistant acid phosphatase (TRAP) staining [52,68]. Lastly, one study conducted mechanical tests on femoral regeneration in mice and assessed the impact of DMOG treatment on bone tissue regeneration by analyzing serum levels of VEGF, OCN, and cyclophosphamide (CTX) by western blotting (WB), polymerase chain reaction (PCR), and enzyme linked immunosorbent assay (ELISA) assays [52].

### 3.6. In Vitro and In Vivo Findings

#### 3.6.1. Osteogenesis

In the in vitro study of osteogenesis (Table 2), the ALP assay was utilized to measure the early osteogenic activity of stem cells. Calcium deposition was identified as a crucial process for osteoblast differentiation and maturation. The ARS method was used to evaluate the late osteogenic mineralization capacity of stem cells. Comparisons of untreated stem cells and DMOG-treated stem cells under normoxia revealed that DMOG-treated stem cells exhibited increased secretion of HIF-1α protein [32,43,48,54]. ALP activity was observed to be heightened after 7 or 14 days of osteogenic culture, while ARS demonstrated the formation of pronounced mineralized nodules and a denser mineral layer after 21 days [42,43,54]. Ding et al. found that DMOG-treated rat ADSCs in osteogenic culture from 3 to 21 days displayed a greater increase in the expression of osteogenesis-related genes such as Runx-2, OCN, ALP, and Collagen-I [42]. Liu et al. showed that when rat BMSCs were cultured with DMOG for 48 h, there was an up-regulation in the expression of osteogenic markers including Runx-2, OCN, ALP, and Collagen-I [54]. 

Fourteen studies examined the use of various bioactive scaffolds incorporating DMOG for the osteogenic culture of stem cells [58,59,60,61,62,63,64,65,66,67,68,72,73,74]. Rat BMSCs were seeded on CBC composite scaffolds containing DMOG and collagen, resulting in the up-regulation of the genes Runx-2, ALP, and OCN after 7 days [58]. Zhang et al. developed ZIF-8 scaffolds with DMOG, which led to increased BMSCs ALP activity, elevated secretion of the proteins ALP, Collagen-I, and P-extracellular regulated protein kinases-1/2 (P-ERK-1/2), and enhanced mineralization of the bone extracellular matrix [59]. DMOG loading onto MSN resulted in the up-regulation of the osteogenic markers Runx-2, Collagen-I, OPN, and OCN, and ALP expression in BMSCs after 7 days [60,61,62]. Significantly elevated ALP activity was observed in BMSCs after 7 and 14 days [60,61], along with increased mineral deposits in BMSCs after 21 days [61,62]. Chen et al. successfully loaded DMOG onto a highly elastic wood-derived scaffold, which showed higher ALP activity in mouse BMSCs, up-regulated expression of the genes Runx-2 and Collagen-I, and increased formation of calcium nodules after 21 days [63]. Inoculation of rat BMSCs on MBG composite scaffolds with DMOG significantly promoted cells proliferation and early osteogenesis, along with the up-regulation of the genes ALP, OCN, and OPN [64]. Mouse BMSCs cultured on various scaffolds containing DMOG exhibited enhanced osteogenesis compared to scaffolds cultured without DMOG [65].

Jahangir et al. cultured rat ADSCs osteoblastically on sponge scaffolds containing alginate-gelatin-tricalcium phosphate with DMOG. After 7 and 14 days, they observed a greater degree of calcium deposition, increased ALP activity, and up-regulation of the genes Runx-2, ALP, and OCN [66]. Yegappan et al. applied DMOG-containing nanocomposite hydrogel to rat ADSCs for in vitro culture. They noted increased ALP activity after 7 and 14 days, the formation of a dense calcium deposition layer after 21 days, and up-regulation of the proteins Runx-2, OPN, and Collagen-I after 7 and 21 days [67]. 

Shang et al. cultured human PDLSCs on different scaffolds, and found that the expression of osteogenic markers Runx-2, bone sialoprotein (BSP), OPN, and OCN were up-regulated on scaffolds containing DMOG after 7, 14, and 21 days, with higher ALP activity measured on day 14 [68].

#### 3.6.2. Bone Regeneration

Among the reports in this review, 13 studies investigated the impact of DMOG on bone regeneration in bone defects [42,43,44,45,58,59,61,62,66,68,71,72,73] (Table 3). All studies utilized micro-CT scanning and histological analysis methods to examine bone regeneration. Some studies also employed IHC or IF staining to assess the expression of osteogenic markers in bone defects [42,43,59,61,62,66]. Zhang et al. conducted a study where DMOG-treated rat BMSCs were inoculated on gelatin sponges and transplanted into mandibular defects in aged SD rats after 8 weeks, resulting in up-regulated expression of Runx-2 and OCN, as analyzed by IHC, leading to significant improvement in mandibular defect repair in vivo [43]. Another study involved treating rat BMSCs with DMOG, inoculating them with β-TCP, and implanting them into a model of cranial bone defects in rats after 8 weeks, which showed an increased presence of newly formed bone tissue [44]. The implantation of a DMOG/collagen/CBC composite scaffold into a rabbit model with femoral condylar defects 12 weeks after inoculation demonstrated that DMOG-encapsulated CBC was the most effective in promoting osteogenesis and bone healing [58]. Zhang et al. loaded DMOG/ZIF-8 into sodium alginate hydrogel and implanted it into a rat cranial defect model. Four weeks after surgery, the results demonstrated that the bone defect in the DMOG-containing group exhibited the highest up-regulation of osteogenic-expressed genes; more newly formed bone tissues were observed. DMOG showed up-regulation in the expression of BMP-2, OPN, and OCN when observed by IF. The up-regulation further enhanced formation of new bone [59]. The results of transplanting composite MSN scaffolds loaded with DMOG into a rat cranial defect model revealed that the DMOG scaffolds group exhibited improvements in bone formation after 6, 8, and 12 weeks, with the up-regulated expression of OCN observed by IF [61,62]. Furthermore, the MBG scaffolds loaded with DMOG also demonstrated a promotion of new bone formation in the cranial bone defects of rats [71].

Ding et al. conducted a study where they loaded DMOG-treated rat ADSCs onto hydrogel scaffolds and transplanted them into rat cranial bone defects. After 8 weeks, the results indicated an up-regulation of OCN expression, showing the most optimal osteogenesis [42] when analyzed by IHC. Similarly, Jahangir et al. implanted rat ADSCs onto DMOG composite scaffolds and in a rat cranial bone defect model. Their in vivo observations demonstrated enhanced bone formation after 6 weeks, with strongly positive OCN and Runx-2 expression by IHC [66]. 

Zhang et al. conducted a study where DMOG-treated hiPSCs were implanted onto β-TCP scaffolds in a rat model of cranial bone defects. They used a tricolor sequential fluorescent labeling method to observe mineralized tissues at 2, 4, and 6 weeks post surgery, and assessed bone regeneration at 8 weeks. Their results demonstrated that scaffolds containing DMOG were the most effective in promoting bone formation [45]. Similarly, Shang et al. transplanted DMOG/nSi-PLGA composite fibrous membranes into periodontal defects in rats, and found that it was the most effective in regenerating the odontoblast-ligament-bone complex in periodontal defects at 1, 2, 4, and 8 weeks points post operation [68]. 

The functions and roles of stem cells in the bone defect microenvironment, such as osteogenic, angiogenic, and immunomodulatory roles, are influenced by the cells, biological factors, biochemical information, and mechanical signals they are exposed to. These factors create ecological niches [79]. As mentioned earlier, bone defects are hypoxic environments, and the enhancement of cellular functions after hypoxic preconditioning of stem cells is mediated by HIF factors [31,32,33,34]. Specifically, HIF-1α plays a crucial role in bone tissue formation and promotes the differentiation of stem cells into osteoblasts and vascular cells [35,36,37]. HIF-1α activates its downstream target genes, facilitating the transport of MSCs, bioactive factors, and nutrients through blood vessels, ultimately enhancing bone formation [80]. On the other hand, HIF-1α can directly influence cells’ metabolism to stimulate bone formation [81]. HIF-1α has become a focal point in research on bone tissue regeneration. However, hypoxic preconditioning poses challenges such as ROS production, cellular aging, rapid degradation of HIF-1α in normoxic environment, and potential mutation due to intermittent hypoxia/ normoxia incubation [82]. Lentiviral transfection for gene amplification carries the risk of infection and tumorigenesis, as HIF-1α is involved throughout the cell cycle [36,40,41]. Therefore, stabilizing HIF-1α with drugs may be a more viable approach.

The studies included in this review demonstrated that DMOG-stimulated stem cells increased the secretion of HIF-1α protein under normoxia [32,43,54]. Moreover, ALP activity notably increased after 7 and 14 days of osteogenic culture [42,43,54,59,60,61,62,63,64,67,68,73], with ARS revealing the formation of more pronounced mineralized nodules after 21 days [42,43,54,59,61,62,63,67]. Detection of osteogenesis-related markers like Runx-2, Collagen-I, OPN, OCN, ALP, etc., was carried out using WB or PCR, showing enhanced expression of osteogenesis-related markers between 3 to 21 days. These effects were consistent across experimental conditions, irrespective of the use of a single flat medium or different bioactive scaffolds [42,43,54,58,59,60,61,62,63,64,65,66,67,68]. DMOG enhanced the osteogenic ability of stem cells. In vivo studies assessed bone regeneration at bone defects through micro-CT scanning, histological, IHC, and IF staining analyses. Whether rats or rabbits were utilized to create bone defect models, the DMOG-loaded group displayed enhanced formation of new bone tissue compared to bioactive scaffolds without DMOG loading. Additionally, combining DMOG with other components showed superior synergistic effects in promoting bone regeneration [42,43,44,45,54,58,59,66,68,69]. 

#### 3.6.3. Angiogenesis

Several studies have demonstrated the effect of DMOG on angiogenesis in stem cells. DMOG was found to increase the expression of angiogenic makers such as HIF-1α and VEGF, leading to enhanced tube formation [32,42,43,46,48,54,69,70]. Besides VEGF, the activation of other angiogenic genes like stromal cell-derived factor-1 (SDF-1), basic fibroblast growth factor (bFGF), and placental growth factor (PLGF) were also observed, with up-regulation starting as early as the first day and reaching a stable level by the third day, maintaining high expression until the 14th day [42,44,45,54]. 

Culturing rat BMSCs on DMOG/CBC/collagen composite scaffolds resulted in increased VEGF expression after 7 days [58]. Similarly, BMSCs in DMOG/MSN showed elevated VEGF expression after 7 and 14 days [60]. Mouse BMSCs grown on MBG containing DMOG for seven days exhibited increased secretion of HIF-1α and VEGF, indicating enhanced vasculogenic capacity and BMSCs differentiation [65]. 

DMOG also improved the angiogenic potential of ADSCs in bone tissue regeneration. Rat ADSCs cultured on sponge scaffolds of alginate-gelatin-tricalcium phosphate containing DMOG for three and seven days showed increased VEGF secretion and up-regulation of the angiogenesis-related genes, CD31, KDR, and CD133 [66].

Human PDLSCs cultured on DMOG/nSi-PLGA membranes displayed increased expression of VEGF, CD31, stem cell factor (SCF), and PLGF, after seven days, with tubule formation assays confirming the formation of a greater number of tubes [68]. 

In the studies reviewed, it was observed that DMOG enhanced the angiogenic capacity of HUVECs. HUVECs cultured on DMOG/ZIF-8 displayed increased cells migration, tube formation, and secretion of proteins such as HIF-1α, VEGF-a, and endothelial nitric oxide synthase (eNOS) [59]. Furthermore, HUVECs grown on DMOG/MSN composite scaffolds showed an increase in tube-like structures [61,62]. When HUVECs were cultured on DMOG-modified material for 8 days, the expression levels of the eNOS, CD31, and VEGF genes were significantly up-regulated [63]. Yegappan et al. utilized a DMOG-containing nanocomposite hydrogel on HUVECs and observed enhanced cells migration and a notable improvement in capillary-like structure formation [67]. 

#### 3.6.4. Vascularization

Nine studies investigated the vascularization of bone defects in vivo, primarily utilizing micro-CT scanning, IHC, and IF staining analysis for microvascular perfusion assessment [42,43,44,45,61,62,66,71]. Zhang et al. conducted a study where DMOG-BMSCs-gelatin sponges were transplanted into mandibular defects in aged SD rats, resulting in a significant up-regulation of VEGF and CD31 after 12 weeks [43]. Ding et al. demonstrated increased vascularization at the defect site and strong positive expression of CD 31 after inoculating DMOG-treated BMSCs onto β-TCP and implanting them into rat cranial bone defects for 8 weeks [44]. DMOG/ZIF-8 composite scaffolds transplanted into rat cranial defects showed up-regulation of angiogenesis-related genes after 4 weeks, with IF staining indicating activation of CD31, HIF-1α, and VEGF, suggesting neovascularization [59]. Similarly, in mouse subcutaneous and rat cranial defect models, composite scaffolds containing DMOG/MSN resulted in a more pronounced tube structure and high expression of CD31, HIF-1α, and α-SMA [61]. Ha et al. transplanted composite scaffolds containing DMOG into a rat cranial bone defect models, observing enhanced bone tissue regeneration and detecting the expression of HIF-1α, CD31, and α-SMA through IF staining after 6 weeks, indicating an effective promotion of vascularization [62]. Additionally, MBG scaffolds with DMOG were found to promote neovascularization in rat cranial bone defects [71].

DMOG-treated ADSCs were cultured on hydrogels and transplanted into rat cranial bone defects. After 8 weeks, a significant increase in vascularization and a high level of CD31 expression were observed at the sites of the bone defects [42]. Composite scaffolds consisting of DMOG and ADSCs were implanted into cranial defects in rats, and after 6 weeks, there was a notable expression of KDR [66]. 

Furthermore, Zhang et al. conducted an experiment where DMOG-hiPSCs-β-TCP were implanted into a rat model with critical-size cranial defects. After 8 weeks, the presence of vascularization in the newly formed bone was confirmed through IHC, along with high levels of CD31, VEGF, and HIF-1α expression [45].

Vasculature is vital for bone growth, development, and the regenerative repair processes of bone defects. It plays a key role in delivering nutrients, facilitating cell communications, and eliminating metabolic waste [83,84,85,86]. VEGF is a major factor in promoting vascularization during bone regeneration [87]. Stabilizing HIF-1α can enhance VEGF expression, promoting angiogenesis [32,43,44,45,46,47,53,54,65] and impacting osteogenesis [88,89,90]. The simultaneous stimulation of osteogenesis and angiogenesis yields a more effective bone regeneration outcome compared to individual stimulation [72,87]. Studies suggest that dual delivery of VEGF and BMP-2 is more effective than BMP-2 alone in promoting bone tissue regeneration [87]. HIF-1α can also upregulate other genes related to angiogenesis, including TGF-β, SDF-1, bFGF, Ang-1, PDGF, PLGF, SCF, etc. [44,45]. DMOG treatment has been shown to enhance the expression of angiogenic markers in stem cells [32,42,43,44,45,46,69] and promote proliferation, migration, and tube formation in HUVECs [59,61,62,63,67], thus effectively promoting angiogenesis. Additionally, DMOG enhances the angiogenic capacity of muscle-derived stem cells (MDSPCs) [49] and dental pulp cells (DPCs) [91,92].

In vivo experiments using CT scans and IHC detection of angiogenic markers such as CD31, α-SMA, VEGF, HIF-1α, and KDR, confirmed the positive effects of DMOG on bone regenerative angiogenesis [42,43,44,45,54,59,62,66,71]. Whether rats or rabbits were used to construct the bone defect models, DMOG was shown to improve angiogenesis in bone defects in vivo.

### 3.7. Therapeutic Effects of DMOG on Disease-Induced Bone Loss and Osteonecrosis

Three studies included in this review examined the therapeutic effects of DMOG on bone-related diseases [52,53,54] (Table 4). Peng et al. utilized female C57BL/6 J mice to establish an OVX model, administering DMOG via intragastric administration to study its impact on bone loss induced by OVX. After 4 weeks, elevated levels of VEGF and OCN were observed in the serum of DMOG-treated mice, along with newly formed blood vessels, increased mineralized surface, and higher rates of mineral deposition and bone formation in bone tissues, further analysis revealed no significant difference in serum levels of CTX and TRAP staining cells between DMOG-treated OVX mice and untreated mice [52], suggesting that DMOG alleviated OVX-induced bone loss by enhancing angiogenesis and stimulating bone tissue regeneration rather than inhibiting osteoclast activity in C57 BL/6 J mice. 

Zhu et al. transplanted DMOG-treated rabbits ADSCs into necrotic areas of the rabbit ONFH model, leading to significant improvement in bone regeneration within 4 weeks, as indicated by strong positive HIF-1α and CD31 expression, as well as a notable increase in hem transfusion [53]. Liu et al. established a rat model of alveolar bone loss induced by nasal congestion in OSAHS, then treated the rats with intrabitoneal injections of DMOG once daily. After 2 weeks, results showed increased alveolar bone mineral density and up-regulation of HIF-1α, VEGF, and ALP expressions [54]. 

### 3.8. Role of DMOG on Chondrogenesis

5 cartilage-related studies were conducted [55,75,76,77,78] (Table 5). Focusing on the use of DMOG on human BMSCs in chondrogenic induction cultures, genes expression analysis at days 5, 7, 14, and 21 showed consistent up-regulation of VEGFA, cGMP-dependent protein kinase 1 (PKG1), EGLN, and Sox-9, while chondrogenic hypertrophy genes were suppressed [55]. Alginate hydrogels releasing DMOG led to pig BMSCs up-regulation of type II Collagen (Collagen-II) and aggrecan, and down-regulation of type X Collagen and matrix metalloproteinase-13 (MMP-13). Implantation of these hydrogels in mice resulted in increased collagen deposition [75]. A study by Chen et al. utilized three-dimensional culture system scaffolds containing DMOG. After 4 weeks, the researchers observed increased expression of Sox-9, aggrecan, and Collagen-II, and decreased MMP-13 expression in rabbit BMSCs. Implantation of these scaffolds in mice showed strong positive expression of Collagen-II in cartilage defects, indicating significant cartilage generation induced by DMOG [76]. Ji et al. developed an injectable drug delivery system incorporating DMOG. They conducted chondrogenic cultures using rat BMSCs with this system and observed a significant increase in extracellular matrix (ECM) deposition after 7 days, the up-regulation of Sox-9, aggrecan, and Collagen-II expression, and increased production of glycosaminoglycans sulfates while the Collagen-X was suppressed. Various scaffolds were implanted into rat knee joints with total osteochondral defects. The results at 6, 12, and 18 weeks indicated that the scaffolds containing DMOG achieved optimal cartilage regeneration [77]. 

Chondrogenesis is a crucial process in bone tissue regeneration, involving the transformation into bone tissue through remineralization [93]. This review examines five studies related on chondrogenesis [55,75,76,77,78], with four studies indicating that DMOG can enhance the differentiation of stem cells toward a chondrogenic lineage, suppress the expression of genes associated with chondrogenic hypertrophy, and increase collagen deposition, thereby effectively promoting cartilage regeneration [55,75,76,77]. These results have demonstrated the use of DMOG-enhanced chondrogenicity in bone defects.

### 3.9. Immunomodulatory Effects of DMOG

During tissue injury, immune cells migrate to the wound sites and participate in tissue remodeling. Macrophages transition from the pro-inflammatory M1 phenotype to the anti-inflammatory M2 phenotype as influenced by immunomodulatory factors to regulate tissue regeneration [61]. Studies suggest that M2 macrophages express high levels of PDGF and VEGF to support vasculature formation [94,95]. Of the studies included in this review, 2 reported that DMOG facilitates the transition of macrophages from a pro-inflammatory M1 phenotype to an anti-inflammatory M2 phenotype, promoting early angiogenesis [61,77]. Therefore, DMOG may also be an effective activator of M2 macrophage polarization, which can promote the angiogenesis process and tissue regeneration [61,77,96]. DMOG activation of HIF-1α was involved in host defense against periapical lesions via down-regulation of NF-κB and pro-inflammatory/bone resorptive cytokines [97]. DMOG can also alleviate osteoarthritis through HIF-1α-mediated mitochondrial autophagy [98]. Further studies are required to elucidate the role of HIF in chronic inflammation, including the HIF-1/HIF-2 switch in immunomodulation, altered cells metabolism, and regulation of mitochondrial ROS. 

### 3.10. Controversies in the Promotion of DMOG on Bone Tissue Regeneration

The studies reviewed in this paper suggest that treating stem cells with DMOG may have therapeutic potential for bone regeneration. However, there remains controversy surrounding the promotion of bone regeneration by DMOG [48,72,73,74,78]. Abu-shahba et al. reported that human ADSCs treated with DMOG showed decreased ALP activity after 7 days, down-regulation of the genes ALP, Runx-2, Collagen-I, and inhibited mineralized matrix formation after 14 days [48]. Similarly, Qi et al. found minimal increases in the expression levels of the genes Runx-2, ALP, and Collagen-I in human BMSCs loaded with DMOG-MBG composite scaffolds when compared to a blank scaffold group after 7 days [72]. On the other hand, Zarkesh et al. observed elevated ALP activity, increased calcium deposition, and up-regulation of osteogenic and angiogenic genes in human BMSCs cultured on CMPs with DMOG after 14 days. However, no significant differences were found in CMPs with or without DMOG [73]. Joseph et al. discovered that the expression of osteogenesis-related genes decreased in scaffolds with DMOG after 21 days of culture with human BMSCs [74]. Lastly, Falconet al. found that DMOG inhibited cartilage formation and negatively impacted the mechanical properties of engineered cartilage when chondrocytes and SYN-MSCs were cultured on particulate-engineered scaffolds containing DMOG for 4 and 6 weeks [78].

The impact of stem cells interacting with scaffolds can vary under different culture conditions, such as in vivo or in the presence of endothelial cells [99]. Previous studies have shown that DMOG induction of stem cells leads to increased production of VEGF, potentially enhancing feedback signaling with endothelial cells. Future in vivo experiments are needed to confirm these findings. It is worth noting that the scaffolds used in these studies did not include growth factors like vitamin D [100] or bone morphogenic protein (BMP) [88], which may act synergistically on bone regeneration in conjunction with DMOG [15,16]. Various factors such as the heterogeneity of stem cell sources, DMOG concentration, cell culture conditions, and types of scaffolds can influence experimental outcomes. Future studies should focus on improving the reporting of results and advocating for quantitative analysis using standardized methods to assess in vivo bone regeneration. 

### 3.11. Secondary Results

Nine studies reported secondary results, primarily focused on signaling pathways [45,52,59,61,62,63,68,75,76] (Table 6), particularly highlighting the role of DMOG as a PHD inhibitor in stabilizing cellular HIF-1α level even in normoxic condition. This stabilization of HIF-1α leads to the up-regulation VEGF expression and secretion, creating a synergistic relationship with osteogenic vascular differentiation [87]. Notably, a positive feedback loop between VEGF and Runx-2 has been observed, where VEGF stimulates osteogenic differentiation and Runx-2 induces VEGF transcription [89,101,102,103,104]. Multiple studies have shown that DMOG activates the phosphatidylinositol 3 kinase/protein kinase B (PI3K/Akt) signaling pathway [45,59,61,62]. With BMSCs cultured on DMOG-containing scaffolds exhibiting significantly higher levels of p-Akt and HIF-1α proteins, the inhibition of PI3K/Akt with LY294002 resulted in a decreasing expression of p-Akt, HIF-1α, and VEGF [45]. Furthermore, transplantation of DMOG/ZIF-8 scaffolds into defects led to overexpression of angiogenesis- and osteogenesis-related genes and pathways, with a notable emphasis on the PI3K/Akt and mitogen-activated protein kinase (MAPK) pathways [59]. Activation of the YAP/TAZ signaling pathway by DMOG through up-regulation of Runx-2 effectively stimulates osteogenic differentiation [63]. Activation of the Wnt/β-catenin signaling pathway has been shown to enhance cellular osteogenic differentiation and promote bone tissue formation [52]. Furthermore, DMOG has been found to modulate the S-mad pathway [75] and the HIF-1α/YAP signaling pathway [76], effectively improving cartilage regeneration.

In terms of selection bias, 12 studies were identified as having subgroup assignment randomization [42,44,45,52,53,54,58,66,68,72,76,77] and were assigned as low risk, but did not provide details about randomization process. Six studies did not describe the baseline characteristics of the animals leading to an unclear risk classification [42,44,59,61,62,75]. None of the included studies mentioned allocation concealment, resulting in an unclear risk classification across the board. 

The main limitations of the study stem from the lack of clarity in various domains assessed by the SYRCLE tool. Insufficient information was provided on experimental testing, phenomena, and attrition bias (randomized housing, study blinding, randomized outcome assessment, outcome blinding, and incomplete outcome data). The study’s strength of evidence is compromised by inadequate reporting of assessment methods, such as specific techniques and their evaluation(Figure 2).

Other biases identified included a lack of clear information on treatment distributional deficiencies, animals, and the presence of a blank group. For instance, some studies did not specify whether a blank group was a separate group or the contralateral limb, or if different limbs of an animal received multiple treatment.

Furthermore, 8 of the studies reviewed require further evaluation for bone regeneration in vivo [48,60,63,64,65,67,74,78].

### 3.12. Prospect and Deficiency

The bone defect presents a hostile environment characterized by ischemia and hypoxia, causing transplanted stem cells to perish quickly. Therefore, it is essential to explore strategies to improve the tolerance of stem cells to this harsh environment for effective bone tissue regeneration. Researches have demonstrated that DMOG can upregulate stemness-related genes such as Kruppel-like factor 4 (KLF4), nanog homology box (NANOG) transcription factor, and recombinant octamer binding transcription factor 4 (OCT4), which play a vital role in maintaining cell stemness, enhancing cells regeneration potential [48], improving the survival of stem cells in harsh environments [42,43,47,48,105,106,107], and promoting cells proliferation [85]. It also boosts the expression of the key homing factor SDF-1 and chemokine receptor type 4 (CXCR-4) [31,33]. These findings suggest that DMOG may have a significant role in promoting the repair of damaged tissues through homing mechanisms.

This review discusses the role of DMOG in osteogenesis and bone tissue regeneration. Other HIF-1α stabilizers such as deferoxamine (DFO) and cobalt chloride (CoCl_2_) have also been studied in the field of bone regeneration. Treating ADSCs with DFO led to up-regulation of HIF-1α, VEGF, and Ang-1 expressions [108]. A study on rabbits showed that local injection of DFO improved vascularization in hormone-induced osteonecrosis of the femoral head [109]. Transplanting DFO-loaded polylactic acid-hydroxy acetic acid scaffolds into rat femoral defects increased bone volume fraction, trabeculae number, and thickness, and reduced trabeculae separation. It also significantly enhanced blood vessel volume and endothelial cells formation [110]. CoCl_2_ was found to enhance osteogenic differentiation of PDLSCs by activating HIF-1α [111]. Furthermore, CoCl_2_ increased HIF expression, stimulated STAT3 phosphorylation, promoted bone formation, and accelerated healing in a mouse femoral defect model [112]. Comparative studies on the effects of these HIF stabilizers in osteogenesis and bone tissue regeneration are needed, along with further exploration of their distinct pathways of action. 

The studies reviewed approached the topic of the impact of DMOG on stem cells differentiation potential and bone tissue regeneration from various perspectives, including: (1) different methods of stem cells extraction and culture, (2) well-characterized stem cells, (3) in vitro assays with concise and complete results, (4) comprehensive reporting of vascularized bone regeneration in vivo using micro CT tomography, histology, IHC, and IF, and (5) complementary analysis and signaling pathways and gene expression. However, it is worth noting that most of the in vitro studies focused on osteogenic induction culture of single cells in a planar medium, which may not fully reflect the complex three-dimensional nature of bone remodeling. Bone remodeling involves a multitude of mechanisms [107], such as cell-to-cell interactions [113,114,115,116], cells-to-surrounding environment interactions [117,118], and interactions with hormones and various biological factors [119,120]. The current culture environment does not accurately mimic the bone defect environment. Additionally, in vivo studies have only been conducted on small animals with different bone defect models, and the healing cycle and mechanical properties of osteogenesis in these animals may differ from those in humans [121]. Presented with the complexity of bone remodeling, further research is needed to evaluate bone regeneration in large animals with severe bone defects or non-healing models. 

Future studies should critically analyze optimal modulation methods based on study objectives, including the appropriate cells source, validation of DMOG’s clinical safety, economically feasible scaffolds for regeneration, and therapeutic effects on bone tissue defects in large animals. Furthermore, the biological effects of DMOG through non-HIF pathways also need to be further explored.

## 4. Conclusions

DMOG has shown potential in maintaining HIF stability, promoting osteogenesis and angiogenesis of stem cells, improving bone tissue regeneration and healing in bone defects, and potentially treating bone-related diseases. Additionally, DMOG has been found to promote chondrogenesis, modulate immune responses, enhance cells tolerance in harsh environments, and mediate migration homing effects at defect sites. DMOG holds promise as a therapeutic tool in bone tissue regeneration engineering. The best methods of using DMOG should be further explored in the future, including the clinical safety of DMOG, the most suitable scaffold for loading DMOG, and in vivo studies of DMOG in large animals and severe bone defects.

## Figures and Tables

**Figure 1 ijms-25-03879-f001:**
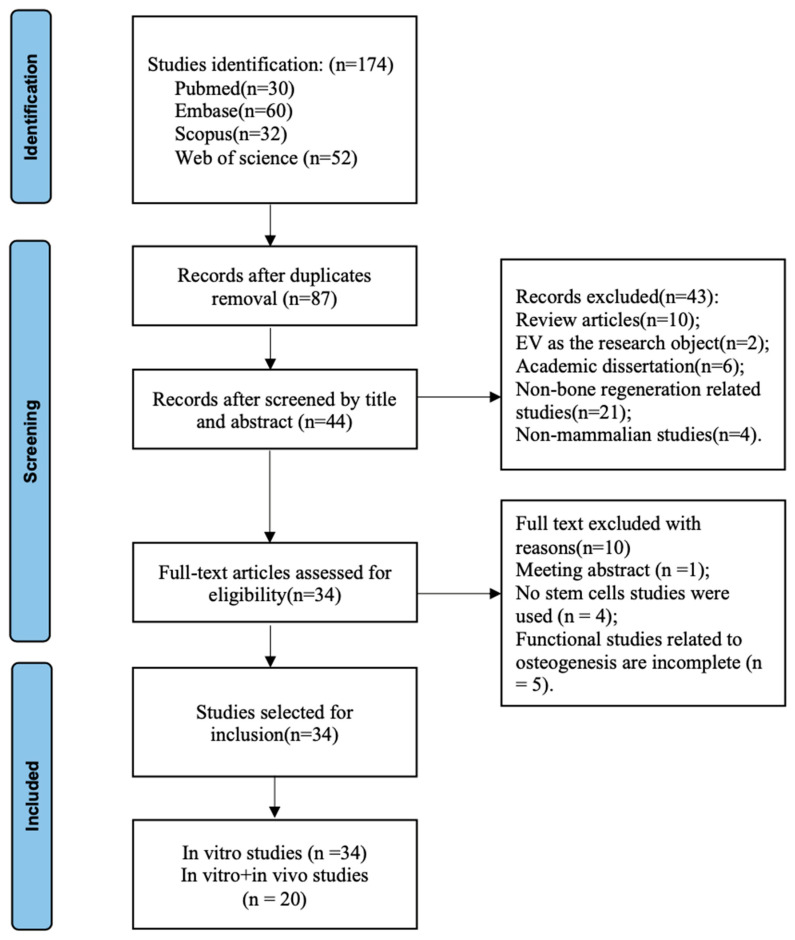
PRISMA flow diagram of literature search and selection process.

**Figure 2 ijms-25-03879-f002:**
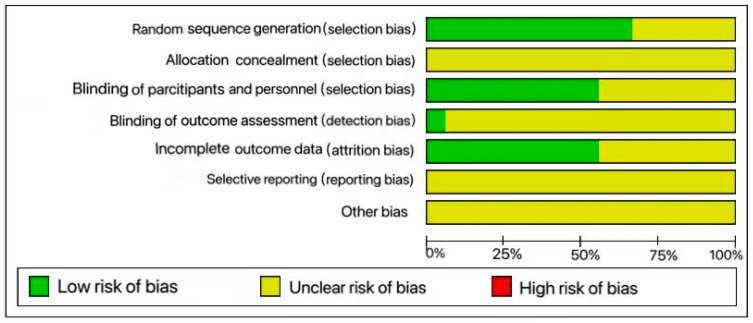
Risk of bias chart for assessing the methodological quality of the 20 papers reporting in vivo studies included in this systematic review.

**Table 1 ijms-25-03879-t001:** MSCs characterization information.

MSCs Source/Origin	Isolation Method	Whether to Identify	Characterization Method	MSCs Makers	Three-Line Differentiation	Reference
**Human BMSCs**	-	YES	Flow cytometry	CD90, CD105, CD73 (+), CD34, CD45 (-)	-	[55]
**Human BMSCs**	-	NO	-	-	-	[60]
**Human BMSCs**	Density gradient centrifugation	NO	-	-	-	[65]
**Human BMSCs**	Enzyme digestion of tissue mass	YES	Flow cytometry	CD73, CD90, CD105 (+), CD80, HLA-DR, CD14, CD34, CD45 (-)	-	[31]
**Human BMSCs**	-	YES	Flow cytometry, identification of three-line differentiation	CD73, CD90, CD106 (+), CD34, CD45 (-)	Osteogenesis, Adipogenesis, Chondrogenesis	[70]
**Human BMSCs**	-	NO	-	-	-	[72]
**Human BMSCs**	-	NO	-	-	-	[73]
**Human BMSCs**	-	NO	-	-	-	[74]
**Rat** **BMSCs**	Enzyme digestion of tissue mass	NO	-	-	-	[43]
**Rat BMSCs**	-	NO	-	-	-	[58]
**Rat** **BMSCs**	Enzyme digestion of tissue mass	NO	-	-	-	[44]
**Rat** **BMSCs**	-	NO	-	-	-	[64]
**Rat** **BMSCs**	-	NO	-	-	-	[77]
**Rat** **BMSCs**	Enzyme digestion of tissue mass	YES	Flow cytometry	CD90 (+), CD34, CD45 (-)	-	[47]
**Rat** **BMSCs**	Lysate separation	NO	-	-	-	[32]
**Mouse BMSCs**	-	NO	-	-	-	[63]
**Rabbit BMSCs**	Density gradient centrifugation	NO	-	-	-	[76]
**Pig BMSCs**	Enzyme digestion of tissue mass	NO	-	-	-	[75]
**Rat** **ADSCs**	Enzyme digestion of tissue mass	YES	Flow cytometry, identification of three-line differentiation	CD29, CD44, CD90 (+), CD31, CD34, CD35 (-)	Osteogenesis Adipogenesis Chondrogenesis	[42]
**Rat** **ADSCs**	Enzyme digestion of tissue mass	YES	Identification of three-line differentiation	-	Osteogenesis Adipogenesis Chondrogenesis	[66]
**Rat** **ADSCs**	-	NO	-	-	-	[67]
**Rabbit ADSCs**	Enzyme digestion of tissue mass	NO	-	-	-	[53]
**Human ADSCs**	Enzymatic digestion and mechanical treatment	YES	Flow cytometry, identification of three-line differentiation	CD73, CD90, CD105 (+), CD14, CD19, CD45, HLA-DR (-)	Osteogenesis Adipogenesis Chondrogenesis	[48]
**Human ADSCs**	Enzyme digestion of tissue mass	NO	-	-	-	[69]
**Dog** **ADSCs**	Enzyme digestion of tissue mass	YES	Flow cytometry, identification of three-line differentiation	CD29, CD73 (+), CD34, CD45 (-)	Osteogenesis Adipogenesis Chondrogenesis	[46]
**Human PDLSCs**	Enzyme digestion of tissue mass	YES	Flow cytometry	CD29, CD44, CD90 (+), CD34, CD45 (-)	-	[68]
**Human** **iPSCs**	-	-	-	CD73, CD90, CD105 (+), CD34, CD45, HLA-DR (-)	Osteogenesis Adipogenesis Chondrogenesis	[45]

-: Inclusion studies are not described; (+): positive expression; (-): negative expression.

**Table 2 ijms-25-03879-t002:** Summary of in vitro methods and outcome.

Stem Cells Conditioning Method	DMOG Concentration	Scaffold	Treatment/Groups	Analysis	Outcome	Ref.
**Rat ADSCs conditioned with DMOG in different concentrations under regular medium for 24 h and then using osteogenic medium for 1, 3, 7, 14, 21 days**	200, 500 and 1000 μM	-	0 μM, 200 μM, 500 μM, and 1000 μM DMOG	WB, qRT-PCR, ALP, ARS	HIF-1α, VEGF protein secretion ↑, Runx-2, OCN, ALP, Collagen-I gene expression ↑, ALP activity ↑ by DMOG, 1000 μM max.	[42]
**Rat ADSCs conditioned with DMOG under regular medium for 48 h and then using osteogenic medium for 7** **,** **21 days**	500 μM	-	Control group, DMOG group	ALP, ARS, WB, tube formation	HIF-1α, VEGF protein secretion ↑, ALP activity ↑, mineralized nodule ↑, increased tubule formation by DMOG.	[43]
**Human ADSCs conditioned with DMOG using osteogenic medium for 7, 14 days**	100, 200, 500 μM	-	Control group, 500 μM DMOG group, 200 μM Baicalein group	WB, qRT-PCR, ALP, ARS	HIF-1α, VEGF protein secretion ↑, ALP activity ↓, mineralized nodule ↓, ALP, BMP-2, Runx-2, Collagen-I gene expression ↓ by DMOG.	[48]
**Rat BMSCs was cultured with DMOG using osteogenic medium for 7, 21 days.**	100, 500, 1000, 2000 μM	-	0 μM,500 μM DMOG	WB, qRT-PCR, ALP, ARS	HIF-1α, VEGF, Runx-2, OCN, ALP, Collagen-I protein secretion ↑, HIF-1α, VEGF, Runx-2, OCN, ALP, Collagen-I gene expression ↑, ALP activity ↑, mineralized nodule ↑ by DMOG.	[54]
**Rat BMSCs was cultured on different scaffolds using osteogenic medium for 7 days**	4 M	CBC	CBC group, Collagen/CBC group and DMOG/Collagen/CBC group	qRT-PCR	ALP, Runx-2, OCN, VEGF gene expression ↑ by DMOG.	[58]
**BMSCs was cultured on different scaffolds using osteogenic medium for 7, 21 days**	-	ZIF8	Control group ZIF8 group ZIF8/DMOG group	WB, ALP, ARS, tube formation	ALP, Collagen-I, P-ERK-1/2, HIF-1α, VEGF-a, eNOS protein secretion ↑, ALP activity ↑ mineralized nodule ↑, increased tubule formation by DMOG.	[59]
**Human BMSCs was cultured on different scaffolds using osteogenic medium for 7, 14 days**	35 μg/mL	MSN	MSN group, DMOG/MSN group	WB, qRT-PCR, ALP	OCN, Runx-2, VEGF protein secretion ↑, Runx-2, OCN, OPN, VEGF gene expression ↑, ALP activity ↑ by DMOG.	[60]
**BMSCs was cultured on different scaffolds** **using osteogenic medium for 7, 14, 21 days**	-	MSN, PLGA, PCL	Control group, PLGA-PCL group, DMOG-MSN/PLGA-PCL group	WB, qRT-PCR, ALP, ARS, tube formation	Runx-2, Collagen-I, OPN protein secretion ↑, Runx-2, ALP, OPN, VEGF, bFGF gene expression ↑, ALP activity ↑, mineralized nodule ↑, increased tubule formation by DMOG.	[61]
**BMSCs was cultured on different scaffolds using osteogenic medium for 7, 14, 21 days**	-	GP, MSN	Control group, GP group, and DMOG-MSN/GP group	WB, qRT-PCR, ALP, ARS, tube formation	Runx-2, Collagen-I, OPN, OCN, HIF-1α, KDR, eNOS, VEGF gene expression ↑, ALP activity ↑, mineralized nodule, increased tubule formation by DMOG.	[62]
**Mouse BMSCs was cultured on different scaffolds using osteogenic medium for 7 days**	-	MBG	MBG group, DMOG-MBG group	WB, qRT-PCR	HIF-1α, VEGF protein secretion ↑, ALP, OPN, OCN gene expression ↑ by DMOG.	[65]
**Rat ADSCs was cultured on different scaffolds using osteogenic medium for 7, 14 days**	-	Sodium alginate-gelatin-β-tricalcium phosphate	Control group, Scaffolds group, Scaffolds-DMOG group	WB, qRT-PCR	VEGF protein secretion ↑, ALP, OCN, Runx2, CD31, KDR, CD133 gene expression ↑ by DMOG.	[66]
**Rat ADSCs was cultured on different scaffolds using osteogenic medium for 7, 14, 21 days**	1 mM	carrageenan nanocomposite hydrogel	Control group, hydrogel group, DMOG-hydrogel group	ALP, ARS, IHC, tube formation	ALP, OPN, Collagen-I protein secretion ↑, ALP activity ↑, mineralized nodule ↑, increased tubule formation by DMOG.	[67]
**Human PDLSCs was cultured on different scaffolds using osteogenic medium for 7, 14, 21 days**	-	PLGA	PLGA, DMOG-PLGA	ALP, qRT-PCR, tube formation	Runx-2, BSP, OPN, OCN, VEGF, CD31, SCF, PLGF gene expression ↑, ALP activity ↑, increased tubule formation by DMOG.	[68]
**Rat BMSCs conditioned with DMOG using regular medium for 1, 3, 7, 14, 21 days**	200, 500 and 1000 μM	-	0 μM, 200 μM, 500 μM, and 1000 μM DMOG	WB, qRT-PCR, ELISA	HIF-1α, VEGF, SDF-1, PLGF, bFGF protein secretion ↑, VEGF, SDF-1, PLGF, bFGF gene expression ↑ by DMOG.	[44]
**Human iPSCs conditioned with DMOG using regular medium for 3, 7 days**	1000 μM	-	0 μM,1000 μM DMOG	WB, qRT-PCR, tube formation	HIF-1α,VEGF protein secretion↑,VEGF, SDF-1, PLGF, bFGF gene expression ↑, increased tubule formation by DMOG.	[45]
**Dog ADSCs conditioned with DMOG using regular medium for 12, 24, 72 h**	0.1, 0.5 mM	-	0 mM, 0.1 mM, 0.5 mM DMOG	WB, qRT-PCR, ELISA, tube formation	HIF-1α,VEGF protein secretion ↑,VEGF gene expression ↑ last for 72 h, bFGF gene expression ↑ at 48 h, but ↓ at 72 h, HGF gene expression ↑ at 6 h, but ↓ at 12 h, Ang-1 ↓ by DMOG, increased tubule formation by DMOG.	[46]
**Human ADSCs conditioned with regular medium for 3, 6, 9 days.** **Human ADSCs conditioned with NPSNPs for 4, 7 days**	50, 100 and 500 μM	NPSNPs	0 μM, 100 μM and 500 μM Control group, NPSNPs, 50 μM DMOG NPSNPs, and 100 μM DMOG NPSNPs	ELISA, tube formation	VEGF protein secretion ↑, increased tubule formation by DMOG.	[69]
**The direct coculture method at ratio 1:1 was used to cultivate human BMSCs and HUVECs together with osteogenic medium for 2, 9 days**	0.5 mM	-	Control group, DMOG group	ELISA, qPCR	VEGF protein secretion ↑, VEGF gene expression ↑ by DMOG.	[70]
**Human BMSCs conditioned with different scaffolds using osteogenic medium for 7 days**	1 mg/mL	PHMG	PHMG group, PHMG-DMOG group, PHMG-BMP2 group, PHMG-DMOG-BMP2 group	WB, qRT-PCR	HIF-1α, VEGF protein secretion ↑, HIF-1α, VEGF gene expression ↑, ALP, Collagen-I, Runx-2 gene expression no up-regulation, ALP, Collagen-I, Runx-2 gene expression ↑ by DMOG is used in conjunction with BMP-2.	[72]
**Human BMSCs conditioned with different scaffolds for 1, 7 and 14 days**	5 mg/mL	CMPs	Mesosphere group, GMP-mesosphere group, CMPs-mesosphere group, DMOG-CMPs-mesosphere group	qRT-PCR, ALP, IHC, histology	Runx2, Sox9, OSX, ALP, OCN, OPN, VEGF, KDR gene expression ↑, IHC for Collagen-I, OCN, VEGF ↑, ALP activity ↑, in CMPs group and DMOG-CMPs group, there was no significant difference between the two groups.	[73]
**Human BMSCs conditioned with different scaffolds for 7, 14 and 21 days**	0.001, 0.01, 0.1 mole	Ca-Sr-MOFs	Ca-Sr-MOFs group, DMOG-Ca-Sr-MOFs group, Osteogenic medium group	qRT-PCR	Runx-2, OCN, Collagen-I gene expression ↓ by DMOG.	[74]

-: Inclusion studies are not described; up-regulation ↑, down-regulation ↓.

**Table 3 ijms-25-03879-t003:** Summary of in vivo methods and outcomes.

Animal Model		Bone Defect Model	Scaffold/Vehicle	DMOG Concen-tration	Treatment Groups	Time Point	Analysis	Outcome	Ref.
**Sprague Dawley (male, 21-month-old, 500–600 g)**		One 5 mm diameter right mandible defects	GS	0.5 mM	GS group, GS-BMSCs group, GS-DMOG-BMSCs (BMSCs were preconditioned with DMOG for 48 h)	8 and 12 weeks	micro CT, histology, and IHC	BV and BV/TV ratio ↑, IHC for OCN, Runx-2, CD31 VEGF ↑, new bone formation by DMOG.	[43]
**Rat**		Two 5 mm diameter calvarial defects	β-TCP scaffolds	1000 μM	β-TCP group, β-TCP-BMSCs group, β-TCP-DMOG-BMSCs group (BMSCs were pretreated with DMOG for 72 h)	8 weeks	micro CT, microfilm perfusion, histology and IHC	BV/TV ratio ↑, IHC for CD31 ↑, new bone and vessel formation by DMOG.	[44]
**New Zealand white rabbits (male, 6 months old, 2.5–3 kg)**		One 5 mm diameter and 8 mm depth critical-sized condyle defect	Collagen-CBC	4 M	Collagen-CBC group, DMOG-Collagen-CBC group	12 weeks	micro CT, histology and IHC	BV/TV ratio ↑, Tb.Sp ↓, new bone formation by DMOG. There was no difference in expression of CD31 and Runx-2 between the with or without DMOG groups.	[58]
**Rat**		A critical-sized cranial defect	ZIF-8, SA hydrogel	-	Control group, ZIF-SA group, ZIF-DMOG-SA group	2 and 4 weeks	micro CT, histology and IF	BV/TV ratio ↑, Tb.n ↓, Tb.Th ↑, IF for BMP-2, OPN, OCN, CD31, HIF-1α, VEGF-a ↑, new bone formation by DMOG.	[59]
**Rat**		Calvarial defects	MSN, PLGA, PCL	-	Control group, PLGA-PCL group, DMOG-MSN/PLGA-PCL group, SrHA/PLGA-PCL group, and DMOG-MSN/SrHA/PLGA-PCL group	4, 8 and 12 weeks	micro CT, microfilm perfusion, histology and IF	BMD ↑, BV/TV ratio ↑, IF for CD31, HIF-1α and OCN ↑, new bone and vessel formation by DMOG and Sr ion. DMOG and Sr ion have synergistic effect.	[61]
**Rat**		Calvarial defects	GP, MSN	-	Control group, GP group, DMOG-MSN/GP group, BFP1-MSN/GP group, DB-MSN/GP group	4, 6 and 12 weeks	micro CT, microfilm perfusion, histology and IF	BV/TV ratio ↑, IF for CD31, HIF-1α, α-SMA and OCN ↑, new bone and vessel formation by DMOG and BFP-1. DMOG and BFP-1 have synergistic effect.	[62]
	**Sprague Dawley rats (male, adult, 250–300 g)**	Two 5 mm diameter calvarial defects	β-TCP scaffolds	1000 μM	β-TCP group, β-TCP-hiPSC-MSCs group, and β-TCP with DMOG-hiPSC-MSCs group (hiPSCs were treated with DMOG for 72 h)	2, 4, 6 and 8 weeks	micro CT, microfil perfusion, histology, IHC and sequential fluorescent labeling	BV/TV ratio ↑, IHC for CD31, HIF-1α and VEGF ↑, new bone and vessel formation by DMOG.	[45]
	**Sprague Dawley rats (male, adult, 250–300 g)**	Two 5 mm diameter calvarial defects	PHMG	1 mg/mL	PHMG group, PHMG-DMOG group, PHMG-BMP2 group, and PHMG-DMOG-BMP2 group	2, 4, 6 and 8 weeks	micro CT, microfilm perfusion, histology, IHC and sequential fluorescent labeling	BV/TV ratio ↑, BMD ↑, IHC for CD31, OCN ↑, new bone and vessel formation by DMOG-BMP2.	[72]
	**Sprague Dawley rats (7–8 weeks, 250–300 g)**	calvarial defects	CMP	5 mg/mL	Control group, trapper without mesosphere group, trapper loaded with mesosphere group, and trapper loaded with DCMP mesosphere group	8 weeks	micro CT, histology	new bone and vessel formation by DMOG.	[73]

BFP-1: bone forming peptide-1; -: Inclusion studies are not described; ↑: up-regulation; ↓: down-regulation.

**Table 4 ijms-25-03879-t004:** Summary of bone-related disease methods and outcomes.

Animal Model	Bone-Related Disease	Scaffold/Vehicle	Concentration of DMOG	Treatment Groups	Time Point Analysis	Analysis	Outcome	Ref.
**Two-month-old female C57BL/6J mice**	OVX	-	5 mg/kg, 20 mg/kg	Sham group, OVX group, OVX + 5 mg/kg/day DMOG group, and OVX + 20 mg/kg/day DMOG group.	4 weeks	micro CT, microfilm perfusion, mechanical testing, histology, TRAP, fluorochrome labeling, bone histomorphometry, and ELISA	BMD, BV/TV, Tb. Th, Tb. N, ↑. Tb.Sp ↓, new bone and new blood vessels form by DMOG. There was no significant difference in serum levels of CTX and TRAP staining cells between DMOG-treated OVX mice and untreated mice.	[52]
**New Zealand rabbits (weighing 2.5–3 kg and** **aged 2–3 months)**	ONFH	Beaver Nano hydrogel	1000 µM	Controls group, core decompression group, core decompression + ADSCs group and core decompression + DMOG-treated ADSCs group.	4 weeks	micro CT, histology, and IHC	BMD, BV/TV ↑, IHC for HIF-1α, CD31 ↑and new bone and new blood vessels form by DMOG.	[53]
**Wistar rats (3 weeks old, male)**	OSAHS nasal obstructed	-	2 mg	Control group, PBS group, DMOG group	2 weeks	micro CT, histology, IHC, WB, and qRT-PCR	BV/TV ↑ b,IHC, qRT-PCR, WB for HIF-1α, VEGF, ALP, Runx2, OCN, Collagen-I ↑ and new bone and new blood vessels form by DMOG.	[54]

-: Inclusion studies are not described; ↑: up-regulation; ↓: down-regulation.

**Table 5 ijms-25-03879-t005:** Summary of chondrogenesis methods and outcomes.

Stem Cells Conditioning Method	Animal Model	Bone Defect Model	DMOG Concentration	Scaffold	Treatment/Groups	Analysis	Outcome	Ref.
**Human BMSCs conditioned with chondroblast medium for 5** **,** **7** **,** **14** **,** **21 days**	-	-	200 μM	-	Control group, 100 μM CoCl_2_ group, 50 μM DFX group and 200 μM DMOG	qRT-PCR, alcian blue staining, glycosaminoglycan quantification	VEGFA, PKG1, EGLN, Sox-9 gene expression ↑, MMP-13 gene expression ↓ by DMOG, DMOG promotes cartilage formation.	[55]
**Pig BMSCs conditioned in alginate hydrogel with chondroblast medium for 7 days. Different alginate hydrogel implanted defect for 4 and 12 weeks**	Balb/C nude mice	Two subcutaneous pockets (one in the shoulder level and one in the hip level)	2.1, 4.2, 6.3 mg/mL	alginate hydroge-l	0, 2.1, 4.2, 6.3 mg/mL alginate hydrogel group, alginate hydrogel-DMOG group	qRT-PCR, micro CT,histology, IHC, gl-ycosaminoglycan and collagen content	Collagen-II, aggrecan gene expression ↑, MMP-13 gene expression ↓, IHC for Collagen-II ↑ by DMOG, DMOG promotes cartilage formation.	[75]
**Rabbit BMSCs conditioned in scaffold with chondroblast medium for 4 weeks. Different scaffold implanted defect for 4 and 8 weeks.**	7-week-old male athymic nude mice	Subcutaneous pocket on the back on each side of the incision	100, 200, 500, and 1000 μM	PLGA PLLA	0 μM, 100 μM, 200 μM, 500 μM, and 1000 μM DMOG PLLA group, PLLA/PLGA-DMOG group, PLLA/PLGA-PTHrP group, and PLLA/PLGA-DP group	qRT-PCR, WB, histology, toluidine blue, safranin O-fast green staining and IHC	Sox-9, aggrecan, Collagen-II gene expression ↑, MMP-13 gene expression ↓, Collagen-II, aggrecan protein secretion ↑, IHC for Collagen-II ↑ by DMOG, DMOG promotes cartilage formation.	[76]
**Rat BMSCs conditioned in scaffold with chondroblast medium for 7 days. Different scaffold implanted defect for 6, 12 and 18 weeks.**	Sprague Dawley rats (male, 300 g)	Femoral trochlear defect (2 mm diameter and 2 mm depth)	25, 50 μg/mL DMOG	CS, HPCH	Control group, HPCH group, 25 μg/mL DMOG-HPCP group, 50 μg/mL DMOG-HPCP group Control group, CS-PMS group, HD/CS-PMS group, CSK-PMS group, and HD/ CSK-PMS group	qRT-PCR, WB,histology,toluidine blue, safranin O-fast green staining, IHC and micro CT	Collagen-II, Sox-9 protein secretion ↑, Sox-9, aggrecan, Collagen-II gene expression ↑, Collagen-X gene expression ↓ by DMOG, DMOG promotes cartilage formation.	[77]
**Chondrocytes and SYN-MSCs were cultured in chondrogenic induction ratio of 1:4 on particulate-engineered scaffolds containing DMOG for 4 weeks and 6 weeks**	-	-	200 μM	Particulate-engineered scaffolds	Untreated group, treated DMOG group	histology (alcian blue and H&E staining), biochemistry, spectral imaging compositional analysis, attenuated total reflection spectroscopy, mechanical assessment	DMOG did not translate to overall increased extracellular matrix deposition, and negatively affected the mechanical competency of the engineered cartilage.	[78]

-: Inclusion studies are not described; ↑: up-regulation; ↓: down-regulation.

**Table 6 ijms-25-03879-t006:** Secondary outcomes related to signaling pathway analysis.

Stem Cells Type	Functions/Signaling Pathways	Ref.
**Human iPSCs,** **BMSCs**	PI3K/Akt	[45,61,62]
**BMSCs**	PI3K/Akt, MAPK	[59]
**Mouse BMSCs**	YAP/TAZ	[63,68]
**Murine mesenchymal C3H10T1/2 clone 8 cells**	Wnt/β-catenin	[52]
**Pig BMSCs**	S-mad	[75]
**Rabbit BMSCs**	HIF-1α/YAP	[76]

## Data Availability

Data are available in the article.

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
