# Peer review of "Effect of Dimethyloxalylglycine on Stem Cells Osteogenic Differentiation and Bone Tissue Regeneration—A Systematic Review"

_ijms, 2024, doi:10.3390/ijms25073879_

Round 1
Reviewer 1 Report
Comments and Suggestions for Authors
In general, the present study can be considered for publication. However, the following points should be considered:
1- The role of biopolymers and bioceramics is not mentioned in the introduction. Please read and use the following items.
https://www.sciencedirect.com/science/article/pii/S0928493119308434
2- Also, the role of blood-derived biomaterials and growth factors involved in the bone regeneration process has not been mentioned. Please read and use the following items.
https://onlinelibrary.wiley.com/doi/abs/10.1002/jbm.a.37489
3- One of the sources of stem cells that is of great interest is endometrial stem cells, which have not been mentioned. Studying and using the following items can strengthen the introduction:
https://link.springer.com/article/10.1007/s10924-022-02615-x
4- The necessity of conducting research and its innovations are not well described. Why should the drug be investigated?
5- Are the results of the study related to different cell sources as well as animal research related to different species with scaffolds of different nature (bioceramic and biopolymer) able to be combined in a systematic review? Please discuss it,
6- The effects of the drug used on the immune system and its role in modulating the immune system in improving bone regeneration should be discussed.
7- The present study should be compared with other similar studies
Reviewer 2 Report
Comments and Suggestions for Authors
The manuscript presents a review of the literature on the osteogenic and angiogenic capabilities of DMOG. The manuscript covers recent literature adequately and is well written. I would request them to address the following comments to improve its over all quality.
1- Please take a look at the full form of DMOG as dimethylglyoxyglycine does not appear correct. In literature it has been mostly reported as dimethyloxalylglycine. Moreover, it would be useful to add some text about its nomenclature, chemical structure, and basic chemistry.
2- In table 3, add the number of days of pre-treatment with DMOG before implantation of cells.
3- Section 3.6.3, covers angiogenesis, not angiogenic differentiation. Either rephrase the heading or add literature where DMOG has been shown to support angiogenic differentiation of stem cells.
4- Section 3.6.4, line 399: How were DMOG/MSN loaded on the scaffold? In addition if other studies have reported DMOG loaded scaffolds without cells, it would be useful to know the methods utilized for loading DMOG.
5- In table 4: Where scaffold/vehicle is not mentioned, how was DMOG delivered?
6- Some minor correction:
- In line 169 CD105 is written twice
- In line 453 'conducted' should be replaced with 'included'
- Table 6: Stem cell profiles doesn't sound right, these probably stem cell types.
7- Finally, there are several grammatical errors in the manuscript, please correct them
Comments on the Quality of English Language
There are several grammatical errors that need to be corrected
Reviewer 3 Report
Comments and Suggestions for Authors
The presented review is interesting, but I think some changes should be made.
1. Please give in the introduction part the common sources of harvesting the stem cells
2. Regarding Figure 1, I think the number of the screened records should be 87 not 44, because I think all 87 records have been screened before rejecting the 43 of them
3. The title of Figure 1 should be only "PRISMA flow diagram of literature search and selection process", not "PRISMA flow diagram of literature search and selection process. PRISMA, Preferred. Reporting Items for Systematic Reviews and Meta-Analyses"
4. Line 132 "87 out of 44 studies were screened by title and abstract", Here I think is a mistake
5. Some lines need references: 165, 168, 172,205-221, 255, 272, 287, 291, 312, 374, 388, 390, 432, 433, 442.
6. The part "3.4 In vitro studies" needs references
7. The tables and the text seem to give the same information, I think it would be better to choose between them and leave only one of them
Round 2
Reviewer 1 Report
Comments and Suggestions for Authors
the manuscript is well-revised and could be recommended for publication
Reviewer 3 Report
Comments and Suggestions for Authors
I have no other comments!